# Y-27632 Induces Neurite Outgrowth by Activating the NOX1-Mediated AKT and PAK1 Phosphorylation Cascades in PC12 Cells

**DOI:** 10.3390/ijms21207679

**Published:** 2020-10-16

**Authors:** So Yeong Park, Jeong Mi An, Jeong Taeg Seo, Su Ryeon Seo

**Affiliations:** 1Department of Oral Biology, Yonsei University College of Dentistry, Seoul 03722, Korea; soyeong433@yuhs.ac (S.Y.P.); myforest@hanmail.net (J.M.A.); 2Department of Molecular Bioscience, School of Bioscience and Biotechnology, Kangwon National University, Chuncheon 24341, Korea

**Keywords:** Y-27632, NADPH oxidase, AKT, p21-activated kinase 1, neurite outgrowth

## Abstract

Y-27632 is known as a selective Rho-associated coiled coil-forming kinase (ROCK) inhibitor. Y-27632 has been shown to induce neurite outgrowth in several neuronal cells. However, the precise molecular mechanisms linking neurite outgrowth to Y-27632 are not completely understood. In this study, we examined the ability of Y-27632 to induce neurite outgrowth in PC12 cells and evaluated the signaling cascade. The effect of Y-27632 on the neurite outgrowth was inhibited by reactive oxygen species (ROS) scavengers such as N-acetyl cysteine (NAC) and trolox. Furthermore, Y-27632-induced neurite outgrowth was not triggered by NADPH oxidase 1 (NOX1) knockdown or diphenyleneiodonium (DPI), a NOX inhibitor. Suppression of the Rho-family GTPase Rac1, which is under the negative control of ROCK, with expression of the dominant negative Rac1 mutant (Rac1N17) prevented Y-27632-induced neurite outgrowth. Moreover, the Rac1 inhibitor NSC23766 prevented Y-27632-induced AKT and p21-activated kinase 1 (PAK1) activation. AKT inhibition with MK2206 suppressed Y-27632-induced PAK1 phosphorylation and neurite outgrowth. In conclusion, our results suggest that Rac1/NOX1-dependent ROS generation and subsequent activation of the AKT/PAK1 cascade contribute to Y-27632-induced neurite outgrowth in PC12 cells.

## 1. Introduction

Neurite outgrowth is necessary during the development of the nervous system and axonal regeneration in the injured nervous system [1]. The small GTPase RhoA has been reported to be a critical negative regulator of neurite outgrowth in developing and regenerating neurons through its downstream effector Rho-associated coiled coil-forming kinase (ROCK) [2]. Activation of RhoA/ROCK contributes to actin cytoskeleton changes, leading to growth cone collapse, which suppresses axonal extension [1,2,3]. Previous studies have shown that blocking RhoA/ROCK resulted in axonal regeneration in embryonic and adult rat neurons [4,5]. The ability of nerve growth factor (NGF) to induce neuronal differentiation was accompanied by ROCK1 inhibition [6,7]. Moreover, upregulation of ROCK has been reported to be associated with neural disorders and neural injury [8].

Y-27632 is known as a potent and specific inhibitor of ROCK activity [9,10]. Its affinity for ROCK is over 200 times or 2000 times higher than that for conventional protein kinase C (PKC) or cAMP-dependent protein kinase (PKA), or myosin light-chain kinase (MLCK), respectively [10]. Many reports have suggested that Y-27632-mediated inhibition of the RhoA/ROCK signaling pathways induces neurite outgrowth in various neuronal cell types. For example, Y-27632 modulates neurite extension by changing the microtubule and vinculin distribution in cultured hippocampal neurons [11]. Y-27632 induces neurite outgrowth and neuronal differentiation in neural stem cells by upregulating YAP expression [12]. In addition, Y-27632 promotes the survival and neurite outgrowth of early postnatal cultured rat retinal neurocytes [13]. Y-27632 also induces an increase in neurite outgrowth in human NT2 model neurons [14]. In PC12 cells, Y-27632 has been reported to potentiate nerve growth factor (NGF)-induced neurite outgrowth [7]. Many studies have suggested that Y-27632 has potential as a therapeutic drug for many neurological disorders, such as spinal cord injury, Alzheimer’s disease (AD), neuropathic pain and neuropsychiatric disorders [8,15,16,17]. For example, the local application of Y-27632 improved functional recovery in mice with spinal cord injured models [8,18]. The use of Y-27632 in animal AD models was efficient in lowering the amount of the toxic Aβ42 levels, but not in total Aβ levels [19]. Consistent with these suggestions, the application of Y-27632 improved motor performance in amyotrophic lateral sclerosis (ALS) male mice [20]. 

Although Y-27632-mediated promotion of neurite outgrowth has been reported in diverse cell types, the precise molecular mechanisms by which Y-27632 induces neurite outgrowth in PC12 cells are not fully understood. In the present study, we investigated the underlying mechanism by which Y-27632 causes neurite outgrowth in PC12 cells and provided biochemical evidence that Y-27632 promotes neurite outgrowth by activating the Rac1/NADPH oxidase (NOX)/AKT/p21-activated kinase 1 (PAK1) signaling cascade. 

## 2. Results

### 2.1. Role of ROS in Y-27632-Induced Neurite Outgrowth in PC12 Cells

We first examined whether Y-27632 induced neurite outgrowth under our experimental conditions. As shown in Figure 1A, Y-27632 effectively induced neurite outgrowth in PC12 cells in a dose- and time-dependent manner. Previously, ROS generation was shown to be involved in the neuritogenesis of PC12 cells [21,22,23]. To test whether the production of ROS contributed to Y-27632-induced neurite outgrowth, we first measured cellular ROS levels using the membrane permeable fluorescent probe DCF-DA in cells that were exposed to Y-27632. As shown in Figure 1B, 25 μM Y-27632 induced time-dependent increases in DCF fluorescence in the cells, indicating that ROS were generated by Y-27632. We next examined whether the prevention of ROS generation with ROS scavengers, such as NAC and trolox, could inhibit Y-27632-induced neurite outgrowth. As shown in Figure 1C, pretreatment of PC12 cells with 5 mM NAC or 500 μM trolox significantly inhibited neurite extension induced by 25 μM Y-27632. The ROS scavenging effects of NAC and trolox were confirmed by the reduced intensity of DCF fluorescence in NAC- and trolox-pretreated cells with Y-27632 (Figure 1D). These results indicate that Y-27632-induced ROS generation is required for the neuritogenesis of PC12 cells.

### 2.2. NOX-Dependent ROS Production during Y-27632-Induced Neurite Outgrowth

Because cellular ROS are mainly generated by NOX complexes [24], we next determined whether the inhibition of NOX prevented Y-27632-induced ROS production and neurite outgrowth. For these experiments, PC12 cells were pretreated with DPI, a NOX-specific inhibitor, prior to Y-27632 administration, and then neurite outgrowth and ROS production were measured (Figure 2). As shown in Figure 2A, NOX1 inhibition with 10 μM DPI significantly blocked neurite outgrowth induced by 25 μM Y-27632. The suppressive effect of DPI on ROS production was confirmed by the decreased level of DCF fluorescence (Figure 2B). We next examined the physiological relevance of NOX in the process of neurite outgrowth in response to Y-27632. For this experiment, we generated stable NOX1-knockdown PC12 cells (Figure 2C). As shown in Figure 2D, 25 μM Y-27632 could not induce neurite outgrowth in NOX1-knockdown cells, indicating that NOX1 is necessary for Y-27632-induced neurite outgrowth. To further confirm the role of NOX1 in neurite outgrowth, NOX1 levels were complemented with exogenous Flag-tagged NOX1 in NOX1-knockdown cells (Figure 2E). To monitor neurite outgrowth, Alexa 633-conjugated phalloidin was used as a specific probe for the binding of filamentous actin. As shown in Figure 2F, NOX1 complementation in NOX1-knockdown cells resulted in neurite outgrowth in response to Y-27632. Collectively, these results indicate that NOX1 is involved in Y-27632-induced neurite outgrowth in PC12 cells. 

### 2.3. Involvement of Rac1-Mediated PAK1 Activation in Y-27632-Induced Neurite Outgrowth

Rac1 is one of the components that activates NOX [25]. Furthermore, Rac1 activity has been shown to be modulated by ROCK1 [26,27]. Therefore, we next assessed whether Y-27632 activated Rac1. To determine whether Y-27632-induced activation of Rac1 was involved in neuritogenesis, PC12 cells were pretreated with various concentrations of NSC23766, a specific inhibitor of Rac1, and then exposed to 25 μM Y-27632 (Figure 3A). The effect of Y-27632 on neurite outgrowth was dose-dependently blocked by NSC23766, indicating that Rac1 activation is involved in Y-27632-induced neurite outgrowth. To further confirm the role of Rac1 in Y-27632-induced neurite outgrowth, PC12 cells were transfected with a dominant negative Rac1 mutant (Rac1N17). As shown in Figure 3B, the expression of dominant negative Rac1 significantly inhibited Y-27632-mediated neurite outgrowth. In contrast, the expression of a constitutive Rac1 mutant (Rac1V12) alone induced neurite outgrowth, confirming that Rac1 is involved in neuritogenesis in PC12 cells. PAK1 was identified as a downstream effector of Rac1 and has been shown to be involved in the growth factor-induced neurite outgrowth in PC12 cells [28,29]. Therefore, we next determined whether PAK1 activation was involved in Y-27632-induced neurite outgrowth. For this experiment, we first measured PAK1 activation in response to Y-27632 using an anti-phospho-PAK1 antibody. As shown in Figure 3C, 25 μM Y-27632 induced a rapid phosphorylation of PAK1. To examine whether PAK1 activation was necessary for Y-27632-induced neuritogenesis, cells were treated with 10 μM IPA-3, a PAK1 inhibitor, followed by 25 μM Y-27632 (Figure 3D). The neuritogenic effect of Y-27632 was blocked by IPA-3 pretreatment, indicating that PAK1 activation is involved in Y-27632-induced neurite outgrowth. The inhibitory effect of IPA-3 on PAK1 activation was confirmed by Western blot analysis (Figure 3E). We next examined whether Rac1 activation affected downstream PAK1 phosphorylation in cells stimulated with Y-27632. PAK1 activation in response to 25 μM Y-27632 was significantly attenuated by the Rac1 inhibitor NSC23766 (Figure 3F), indicating that Rac1 activation is required for PAK1 activation. Taken together, these results suggest that Rac1-mediated PAK1 activation is involved in Y-27632-induced neurite outgrowth in PC12 cells.

### 2.4. Activation of PAK1 Is Induced by NOX-Dependent ROS Generation in Y-27632-Stimulated PC12 Cells

We next examined whether Y-27632-induced ROS generation played a role in PAK1 phosphorylation. As shown in Figure 4A, pretreatment with ROS scavengers, such as 5 mM NAC and 500 μM trolox, significantly attenuated PAK1 phosphorylation induced by 25 μM Y-27632. Consistent with these results, the inhibition of ROS generation with 10 μM DPI, a NOX inhibitor, subsequently inhibited PAK1 phosphorylation (Figure 4B). Furthermore, PAK1 phosphorylation in response to 25 μM Y-27632 was suppressed in NOX1-knockdown cells (Figure 4C). These results indicate that Y-27632-induced ROS generation activates of NOX, which subsequently activates its downstream effector PAK1 in PC12 cells.

### 2.5. Involvement of the AKT-PAK1 Cascade in Y-27632-Induced Neurite Outgrowth

It has been reported that PAK1 is phosphorylated by the activation of the serine/threonine kinase AKT [30]. Because AKT has been suggested to be an important regulator of NGF-induced neurite outgrowth [31,32], we next examined the possible involvement of AKT in Y-27632-induced neurite outgrowth. As shown in Figure 5A, AKT phosphorylation was induced by 25 μM Y-27632 treatment in PC12 cells. In accordance with AKT phosphorylation, 25 μM Y-27632-induced neurite outgrowth was significantly blocked by 5 μM MK2206, an AKT inhibitor (Figure 5B). The MK2206-induced inhibition of AKT phosphorylation was confirmed by measuring the p-AKT protein expression level normalized to that of total AKT in PC12 cells stimulated with 25 μM Y-27632 in the presence or absence of 5 μM MK2206 (Figure 5C). We next examined whether Rac1 contributed to Y-27632-induced AKT phosphorylation. As shown in Figure 5D, 60 μM NSC23766, a Rac1 inhibitor, blocked AKT phosphorylation induced by 25 μM Y-27632, indicating that Rac1 induces subsequent AKT activation leading to the neurite outgrowth. To examine whether Y-27632-induced ROS generation caused the activation of AKT, the effect of ROS scavengers, including NAC and trolox, on AKT phosphorylation was measured. As shown in Figure 5E, scavenging ROS with 5 mM NAC or 500 μM trolox inhibited downstream AKT phosphorylation. Furthermore, NOX inhibition with 10 μM DPI and NOX1 knockdown using shRNA suppressed AKT phosphorylation induced by 25μM Y-27632 (Figure 5F,G). We next examined whether the phosphorylation of PAK1 was attributed to the activation of AKT. As shown in Figure 5H, PAK1 phosphorylation was significantly suppressed by the AKT inhibitor MK2206 (5 μM), indicating that AKT activation is required for the PAK1 activation. Collectively, Y-27632 induces neurite outgrowth by activating the Rac1/NOX1/AKT/PAK1 cascade in PC12 cells (Figure 6).

## 3. Discussion

ROS function as signaling molecules in a variety of biochemical processes, including neurite outgrowth and neuronal differentiation [33,34]. During rat brain development, the modulation of ROS levels influences multiple aspects of neuronal differentiation [33]. Higher levels of ROS are present in newborn neurons and persist in the neurogenic zones in the adult brain [35]. ROS-exposed neural progenitor cells show increased numbers of neurons and oligodendrocytes through increased expression of the proneural gene Ngn2 and neural marker gene β-III tubulin [34]. NGF-mediated neurite outgrowth and neuronal differentiation are accompanied by increased ROS levels in PC12 cells [36]. NOX enzymes are the main source of ROS and have been shown to be involved in growth cone formation and neurite outgrowth [24,36,37]. Treatment of neuronal cells with the NOX inhibitor DPI inhibits NGF-induced ROS production and neurite outgrowth, suggesting that NOX-dependent ROS generation is involved in neuronal differentiation [36]. In accordance with these reports, we found that Y-27632-induced neurite outgrowth was prevented by not only ROS scavengers but also NOX1 and Rac1 inhibitors in PC12 cells. To further evaluate the endogenous role of NOX1 in Y-27632-induced neurite outgrowth, we generated the stable NOX1-knockdown cells using a pool of 3 different target-specific shRNA-expressing plasmids. The effect of Y-27632 on the neurite outgrowth was consistently perturbed in the stable NOX1-knockdown cells. However, further studies will be needed to confirm the effect of NOX1-knockdown on the Y-27632-induced neurite outgrowth in PC12 cells using different shRNAs.

The small GTPases Rac1 and RhoA have been implicated in neuronal differentiation by regulating cytoskeletal rearrangement [38,39]. Rac1 and RhoA have opposing functions in the process of neurite outgrowth [26,40]. For example, the expression of constitutively active Rac1 leads to neurite outgrowth through the generation of filopodia and lamellipodia in the developing growth cone, and dominant negative Rac1 inhibits these effects in N1E-115 neuroblastoma cells [40]. In contrast, activation of RhoA causes neurite retraction and the inhibition of RhoA stimulates neurite outgrowth in the same cells [40]. Consistent with these reports, we found that the inhibition of ROCK1, which is a target of RhoA, with Y-27632 induced neurite outgrowth, and this neurite outgrowth-promoting effect was suppressed by Rac1 inhibition, suggesting that RhoA acts negatively on Rac1 activity in neurite outgrowth in PC12 cells. 

The PAK family of serine/threonine kinases has been identified as downstream effectors of the activated GTPases Rac and Cdc42 [41]. Expression of PAK stimulates morphological changes in the actin cytoskeleton associated with Rac and Cdc42 [41]. Membrane targeting of PAK1 has been shown to drive neurite extension in PC12 cells similar to that induced by NGF [29]. Based on these reports, we found that PAK1 activation is necessary for Y-27632-induced neurite outgrowth and that Rac1 activation is required for Y-27632-induced PAK1 activation.

AKT activation is required for PAK1 activation and modulates cell motility [42,43]. The involvement of the AKT pathway has been widely shown to promote neuronal survival [44,45]. For example, AKT prevents injury-induced motoneuron death and accelerates axonal regeneration [45]. The activity of AKT is increased up to 14-fold in serum-starved PC12 cells in response to NGF [46]. AKT has also been identified as a key regulator of neurite outgrowth in PC12 cells [32]. AKT activation promotes neurite elongation in NGF-stimulated PC12 cells, and constitutively active AKT expression spontaneously induces neurite outgrowth in PC12 cells [32]. In support of these reports, we showed in this study that the AKT inhibitor MK2206 prevented PAK1 phosphorylation and neurite outgrowth caused by Y-27632 in PC12 cells, indicating that AKT exerts its neurite-inducing effect by modulating PAK1 activation. In addition, we observed that the prevention of ROS generation consistently perturbed AKT and subsequent PAK1 activation in response to Y-27632. Our data suggest that AKT is activated by ROS and acts as a mediator linking the downstream effector PAK1 in the Y-27632-induced neurite outgrowth process. In support of these results, ROS have been shown to contribute to AKT activation in various cell types [47,48,49,50]. For example, it was reported that ROS production is required for epidermal growth factor (EGF)-induced AKT activation in mediating tumor angiogenesis in ovarian cancer cells [47]. In addition, NOX1 was shown to promote the self-renewal activity of CD133+ thyroid cancer cells through activation of AKT signaling [50]. ROS-mediated AKT activation is also known to contribute to axonal regeneration and functional recovery after spinal injury [48]. Furthermore, it has been reported that ROS directly activates AKT signaling and indirectly activates AKT signaling by inactivating phosphatase and tensin homolog (PTEN), which inhibit AKT [48,49]. NOX oxidizes PTEN, which leads to its inactivation, thus stimulating AKT signaling and axonal outgrowth [48].

In the present study, we demonstrated that Y-27632, a specific ROCK inhibitor, induces neurite outgrowth through activation of the Rac1/NOX1/AKT/PAK1 pathway in PC12 cells. The molecular targets of Y-27632 are not only significant for understanding how this compound regulates neurite outgrowth in vivo but also relevant for improving the therapeutic efficacy. In support of our notion, Li et al. have recently suggested that the administration of Y-27632 effectively increased the survival rate and behavioral performance of rats from cerebral ischemic injury [51]. Blocking of ROCK with Y-27632 prevented the initiation of neuropathic pain after peripheral nerve injury in mice [52]. Despite these reports, the use of Y-27632 in vivo has been limited because it is metabolized very rapidly after oral administration and its brain penetration was too low to achieve therapeutic levels for the CNS disease. Thus, the intensive investigation will be needed to evaluate the efficacy of Y-27632 before the application of human therapeutics. 

Y-27632 has been extensively studied in a variety of cellular models. In accordance with these reports, our study reported the neurite-promoting effect of Y-27632 and suggested the underlying molecular signaling cascade in PC12 cell lines. Due to the limited studies using an in vitro system, further studies are expected to confirm whether our hypothesis can be applied to the primary neurons and to the diverse in vivo animal models. 

## 4. Materials and Methods 

### 4.1. Cell Culture

The PC12 cells used in this study were purchased from the American Type Culture Collection (ATCC, Manassas, VA, USA). The cells were maintained in RPMI-1640 medium with 10% horse serum (Gibco, Carlsbad, CA, USA), 5% fetal bovine serum (Gibco, Carlsbad, CA, USA) and 100 mg/mL antibiotic-antimycotic reagent (Gibco, Carlsbad, CA, USA) at 37 °C in a 5% CO2 incubator.

### 4.2. Measurement of Neurite Outgrowth

PC12 cells were plated at a density of 2 × 10^5^ cells/well in a 100 ug/mL of collagen (Enzo Biochem, Farmingdale, NY, USA)-coated 6-well culture plate. The ROCK inhibitor Y-27632 was diluted in culture media to the final concentrations indicated in each experiment. A process that was longer than the cell body diameter was defined as a neurite and the images of living cells were captured using a Ti-U (Nikon, Minato, Tokyo, Japan) camera.

### 4.3. Pharmacological Treatment

®-(+)-trans-N-(4-pyridyl)-4-(1-aminoethyl)-cyclohexanecarboxamide dihydrochloride (Y-27632), NSC23766, and IPA-3 were obtained from Tocris Bioscience (Ellisville, MO, USA). Diphenyleneiodonium (DPI), N-acetylcysteine (NAC), and 6-hydroxy- 2, 5, 7, 8-tetramethylchromane-2-carboxylic acid (trolox) were obtained from Sigma-Aldrich (St. Louise, MO, USA). MK-2206 was purchased from Selleckchem (Houston, TX, USA). Y-27632, NSC23755, NAC, and trolox were dissolved in ddH_2_O. IPA-3, DPI, and MK-2206 were dissolved in dimethyl sulfoxide (DMSO). All chemicals were diluted in culture media to the final concentrations just before use. To investigate the effect of each inhibitor, PC12 cells were pretreated with each inhibitor for 30 min prior to Y-27632 treatment. Culture media was used as a control.

### 4.4. Measurement of Intracellular Reactive Oxygen Species (ROS)

ROS were detected by labeling the cells with 1 μM CM-H2DCFDA (Thermo Scientific, Waltham, MA, USA) for 45 min. Cells were seeded in a 12-well plate containing cover slips at a density of 1 × 10^5^/well. Then, the cells were incubated with CM-H2DCFDA dye for 45 min at 37 °C. The fluorescence intensity was measured using a Zeiss LSM 700 confocal microscope (Ex/Em = 495/519 nm) (Zeiss, Oberkochen, BW, Germany).

### 4.5. Generation of Stable NOX1 shRNA-Expressing Cells

To generate a stable cell line, the cells were transfected with a NOX1 shRNA plasmid vector (sc-156079-SH, Santa Cruz, Santa Cruz, CA, USA). After 72 h of transfection, the cells were selected using diluted 10 μg/mL of puromycin (Enzo Biochem, Farmingdale, NY, USA) in the culture media.

### 4.6. Western Blotting

Cells were lysed using RIPA lysis buffer (T&I, Chuncheon, Kangwon, Korea) supplemented with a phosphatase and protease inhibitor cocktail (Roche, Grenzacherstrasse, Basel, Switzerland), and the lysates were centrifuged at 13,000 rpm for 15 min at 4 °C to separate the supernatants. Cell lysates were subjected to SDS-PAGE gels and transferred to PVDF membranes (Merck Millipore, Burlington, MA, USA). The membranes were blocked for 1 h with 5% skim milk (BD Biosciences, Franklin Lakes, NJ, USA), and then the primary antibodies were added and incubated overnight at 4 °C. The blots were detected using anti-phospho-PAK1 (1:1000, Cell signaling, Danvers, MA, USA), anti-PAK1 (1:1000, Santa Cruz, Santa Cruz, CA, USA), anti-phospho-AKT (1:1000, Cell signaling, Danvers, MA, USA), anti-AKT (1:1000, Cell Signaling, Danvers, MA, USA), anti-NOX1 (1:1000, Santa Cruz, Santa Cruz, CA, USA), and anti-GAPDH (1:5000, Santa Cruz, Santa Cruz, CA, USA) antibodies. The specific protein bands were detected using chemiluminescence reagents (GE Healthcare Life Science, Chicago, IL, USA) and quantified using ImageJ software (National Institutes of Health, Bethesda, MD, USA).

### 4.7. Cell Transfection

For transfection, PC12 cells were seeded in 6-well plates at a density of 2 × 10^5^ / well, and then the cells were transfected using Lipofectamine LTX (Thermo Scientific, Waltham, MA, USA) transfection reagent according to the manufacturer’s protocol. We used expression vectors for wild type Rac1 (pDsRed2-Rac1-WT), a dominant negative form of Rac1 (pDsRed2-Rac1-N17), a constitutively active form of Rac1 (pDsRed2-Rac1-V12) or pcDNA3.1-Flag-NOX1, and Lipofectamine LTX to transfect PC12 cells. Rac1 expression vectors were kindly provided by Prof. Jae Hong Kim. (Korea University, Seoul, Korea). After 24 h, we assessed the morphology of the transfected cells with pDsRed2-Rac1-WT, pDsRed2-Rac1-N17, and pDsRed2-Rac1-V12 using a Ti-U camera. NOX1 knockdown cells were rescued by transient transfection with a codon-optimized pcDNA3.1-Flag-NOX1 plasmid. After 24 h, the cells were fixed for immunocytochemistry.

### 4.8. Immunocytochemistry (ICC)

For immunocytochemistry, the cells were treated with 4% paraformaldehyde (PFA) for 10 min for fixation, permeabilized with 0.2% Triton X-100 in PBST for 30 min and blocked with 5% goat serum (Jackson Laboratory, Bar Harbor, ME, USA) in PBS for 1 h. After that, the cells were incubated with rabbit anti-DsRed2 antibody (1:400, Clontech, Mountain View, CA, USA) or rabbit anti-Flag antibody (1:400, Sigma-Aldrich, St. Louise, MO, USA) diluted in 5% goat serum overnight at 4 °C. Then, the cells were incubated with Alexa 488- or 568-conjugated secondary antibodies (1:400, Thermo Scientific, Waltham, CA, USA) for 1 h at room temperature, and Alexa 633-phalloidin (1:800, Thermo Scientific, Waltham, CA, USA) was used to counterstain the actin filaments. The samples were mounted onto slides with Vectashield (Vector lab, Burlingame, CA, USA), and images were analyzed using a Zeiss LSM 700.

### 4.9. Statistical Analysis

All data are presented as the means ± S.E.M using GraphPad Prism statistical software. The data were assessed using Student’s *t*-test for comparisons between two groups. One-way analysis of variance (ANOVA) was used to compare multiple groups. *p*-values less than 0.05 were considered statistically significant.

## Figures and Tables

**Figure 1 ijms-21-07679-f001:**
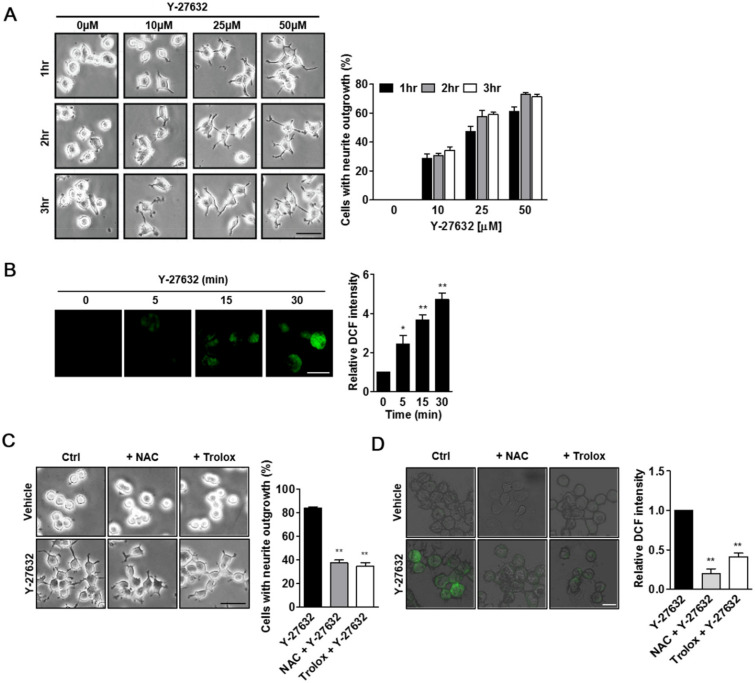
Y-27632 induces neurite outgrowth and ROS production in PC12 cells. (**A**) PC12 cells were exposed to different concentrations of Y-27632 (10–50 μM) for the indicated times. Microphotographs of the cells were taken, and representative images are shown in the left panel. Cells with neurite processes longer than the cell body diameter were counted, and the percentages of neurite-containing cells are shown in the right panel. Note that neurite outgrowth was promoted by Y-27632 in a time- and concentration-dependent manner. Scale bar = 50 µm. (**B**) PC12 cells were incubated with 2 μM DCF-DA and then treated with 25 μM Y-27632 for 30 min. The fluorescence intensity was measured using a confocal microscope. Scale bar = 20 µm. Culture images are at a magnification of 63×. (**C**) PC12 cells were treated with 25 μM Y-27632 for 2 h in the presence or absence of ROS scavengers, such as 5 mM NAC and 500 μM trolox. Cells with neurite processes longer than the cell body diameter were counted, and the percentages of neurite-containing cells were determined. Scale bar = 50 µm. Culture images are at a magnification of 20×. (**D**) PC12 cells were incubated with 2 μM DCF-DA and treated with 5 mM NAC or 500 μM trolox. Cells were then exposed to 25 μM Y-27632 for 1 h. The fluorescence intensity was measured using a confocal microscope. Note that ROS were generated by Y-27632, and ROS scavengers, such as NAC and trolox, prevented ROS production and neurite outgrowth triggered by Y-27632. Scale bar = 20 µm. Culture images are at a magnification of 40×. The data represent the means ± S.E.M. of at least four independent experiments. * *p* < 0.05; ** *p* < 0.01 compared with the control group (vehicle or Y-27632 alone group).

**Figure 2 ijms-21-07679-f002:**
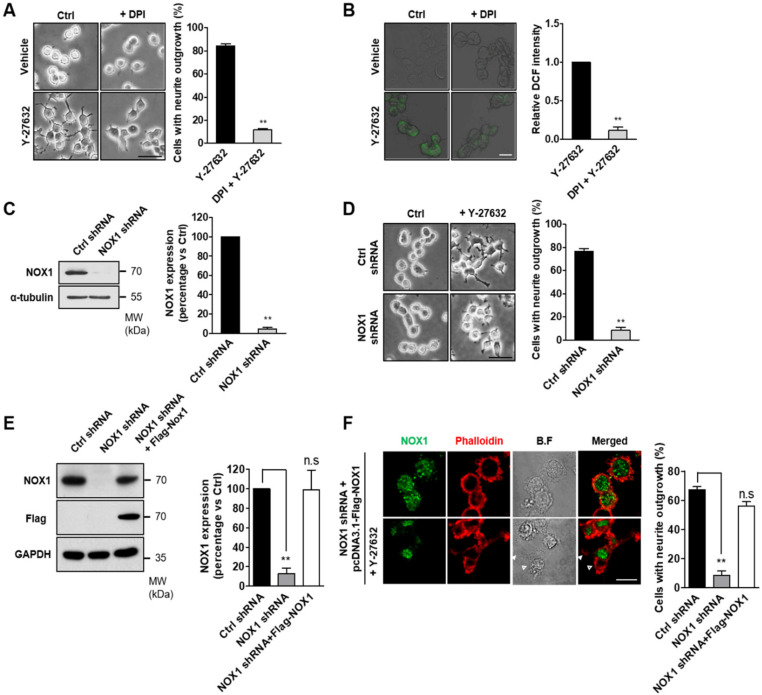
NOX1 is required for ROS generation and neurite outgrowth induced by Y-27632. (**A**) PC12 cells were treated with 25 μM Y-27632 for 2 h in the presence or absence of 10 μM DPI. Microphotographs of the cells were taken, and representative images are shown in the left panel. Cells with neurite processes longer than the cell body diameter were counted, and the percentages of neurite-containing cells are shown in the right panel. Scale bar = 50 µm. Culture images are at a magnification of 20×. (**B**) PC12 cells were incubated with 2 μM DCF-DA and treated with 10 μM DPI. Cells were then exposed to 25 μM Y-27632 for 1 h. The fluorescence intensity was measured using a confocal microscope. Note that the NOX-1 inhibitor DPI prevented both ROS generation and neurite elongation caused by Y-27632. Scale bar = 20 µm. Culture images are at a magnification of 40×. (**C**) Western blot analysis of the lysates of cells transfected with empty vector (control shRNA) or shRNA targeting NOX1 (NOX1 shRNA). Gels were transferred and blotted with specific antibodies against NOX1 or α-tubulin. (**D**) PC12 cells transfected with control shRNA or NOX1 shRNA were treated with 25 μM Y-27632 for 2 h. Cells with neurite processes longer than the cell body diameter were counted, and the percentages of neurite-containing cells were determined. Note that knockdown of NOX1 by shRNA transfection inhibited neurite outgrowth induced by Y-27632. Scale bar = 50 µm. Culture images are at a magnification of 20×. (**E**) Western blot analysis of the lysates of NOX1 knockdown cells transfected with the Flag-NOX1 plasmid for 24 h. Gels were transferred and blotted with specific antibodies against NOX1, Flag or GAPDH. Note that the expression of NOX1 increased after Flag-NOX1 transfection in NOX1-knockdown cells. (**F**) Representative immunocytochemistry analysis of NOX1 (green) and phalloidin (red) in Flag-NOX1 transfected NOX1-knockdown cells treated with 25 μM Y-27632. Cells with neurite processes longer than the cell body diameter were counted, and the percentages of neurite-containing cells are shown in the right panel. Note that Y-27632 promoted neurite outgrowth in NOX1-knockdown cells transfected with the Flag-Nox1 plasmid. Arrowheads indicate a neurite promoted by Y-27632 from a cell transfected with Flag-Nox1. Empty arrowheads indicate a nontransfected cell that does not bear neurites. Scale bar = 20 µm. Culture images are at a magnification of 63×. The results are presented as the means ± S.E.M of at least three independent experiments. ** *p* < 0.01; n.s, not significant.

**Figure 3 ijms-21-07679-f003:**
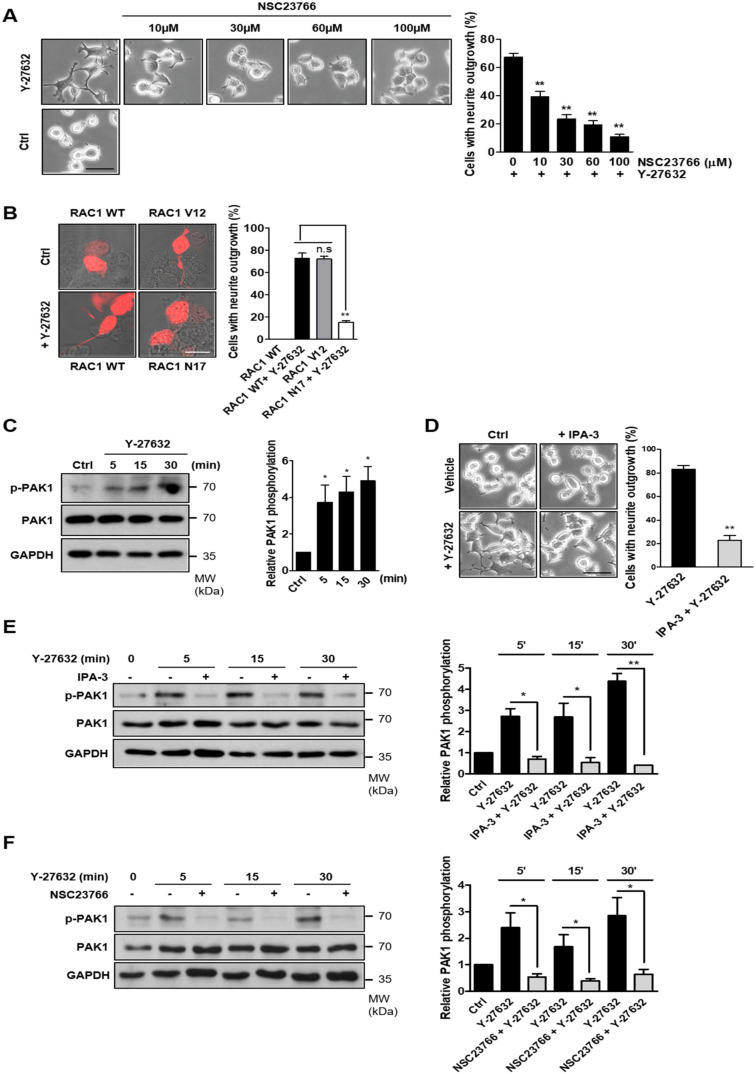
Activation of the Rac1-PAK1 cascade results in neurite outgrowth in PC12 cells. (**A**) PC12 cells were pretreated with different concentrations of NSC23766 (10, 30, 60, or 100 μM), a selective inhibitor of the Rac1-GEF interaction, for 30 min and then exposed to 25 μM Y-27632 for 2 h. Cells were counted by determining the percentage of cells bearing neurites among the total cells. Note that Y-27632-induced neurite outgrowth was inhibited by the Rac1 inhibitor NSC23766 in a dose-dependent manner. Scale bar = 50 µm. Culture images are at a magnification of 20×. (**B**) PC12 cells were incubated with the indicated plasmids (Rac1 WT; wild-type Rac1, Rac1 V12; a constitutively active form of Rac1 or Rac1 N17; a dominant negative form of Rac1) for 24 h. Cells were then exposed to 25 μM Y-27632 for 2 h. The percentage of neurite-bearing cells among DsRed-positive cells was determined. Note that Y-27632 did not induce neurite outgrowth in cells transfected with Rac1 N17, and by contrast, elongated neurites were observed in Rac1 V12-transfected cells. Scale bar = 20 µm. Culture images are at a magnification of 63×. (**C**) PC12 cells were stimulated with 25 μM Y-27632 for the indicated times, and cell lysates were then immunoblotted with antibodies against phosphorylated PAK1 (at Ser199/204), PAK1, and GAPDH. Note that Y-27632 induced phosphorylation of PAK1. (**D**) PC12 cells were stimulated with 25 μM Y-27632 for 2 h in the presence or absence of 10 μM IPA-3, a PAK1 inhibitor. Cells with neurite processes longer than the cell body diameter were counted and the percentages of neurite-containing cells were determined. Note that pretreatment of the cells with IPA-3 inhibited Y-27632-induced neurite outgrowth. Scale bar = 50 µm. Culture images are at a magnification of 20×. (**E**) PC12 cells were stimulated with 25 μM Y-27632 for the indicated times in the presence or absence of 10 μM IPA-3, and cell lysates were then immunoblotted with antibodies against phosphorylated PAK1 (at Ser199/204), PAK1, and GAPDH. Quantification of p-PAK1 protein expression was normalized to that of the total PAK1. (**F**) PC12 cells were stimulated with 25 μM Y-27632 for the indicated times in the presence or absence of 60 μM NSC23766, and cell lysates were then immunoblotted with antibodies against phosphorylated PAK1 (at Ser199/204), PAK1, and GAPDH. Quantification of p-PAK1 protein expression was normalized to that of total PAK1. The data represent the means ± S.E.M. of at least five independent experiments. * *p* < 0.05; ** *p* < 0.01; n.s, not significant.

**Figure 4 ijms-21-07679-f004:**
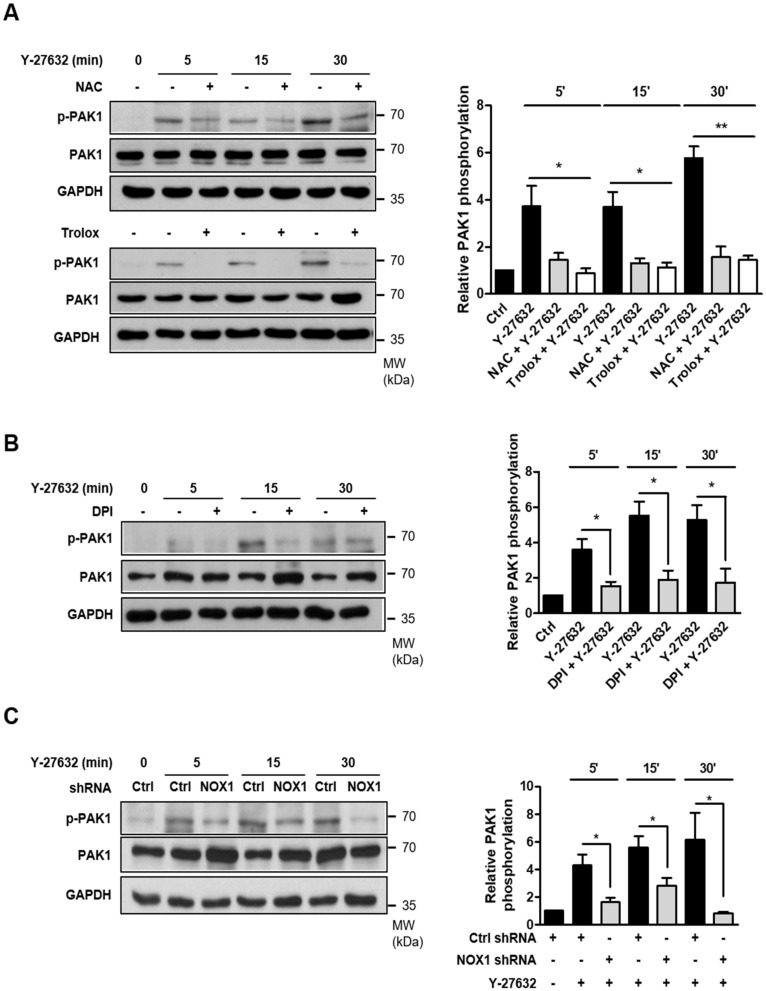
NOX1-mediated ROS generation is required for Y-27632-induced PAK1 activation. (**A**) PC12 cells were treated with 25 μM Y-27632 for the indicated times in the presence or absence of ROS scavengers, such as 5 mM NAC and 500 μM trolox, and cell lysates were then immunoblotted with antibodies against phosphorylated PAK1 (at Ser199/204), PAK1, and GAPDH. The p-PAK1 protein expression level was normalized to that of total PAK1 and is shown in the right panel. (**B**) PC12 cells were treated with 25 μM Y-27632 for the indicated times in the presence or absence of 10 μM DPI, and the cell lysates were then immunoblotted with antibodies against phosphorylated PAK1 (at Ser199/204), PAK1, and GAPDH. The p-PAK1 protein expression level was normalized against PAK1 and is shown in the right panel. (**C**) PC12 cells transfected with control shRNA or NOX1 shRNA were treated with 25 μM Y-27632, and the cell lysates were then immunoblotted with antibodies against phosphorylated PAK1 (at Ser199/204), PAK1, and GAPDH. The p-PAK1 protein expression level was normalized to that of total PAK1 and is shown in the right panel. Note that scavenging ROS with NAC or trolox, inhibiting NOX1 with DPI, and knocking down NOX1 all inhibited Y-27632-induced PAK1 phosphorylation. The data represent the means ± S.E.M. of at least three independent experiments. * *p* < 0.05; ** *p* < 0.01.

**Figure 5 ijms-21-07679-f005:**
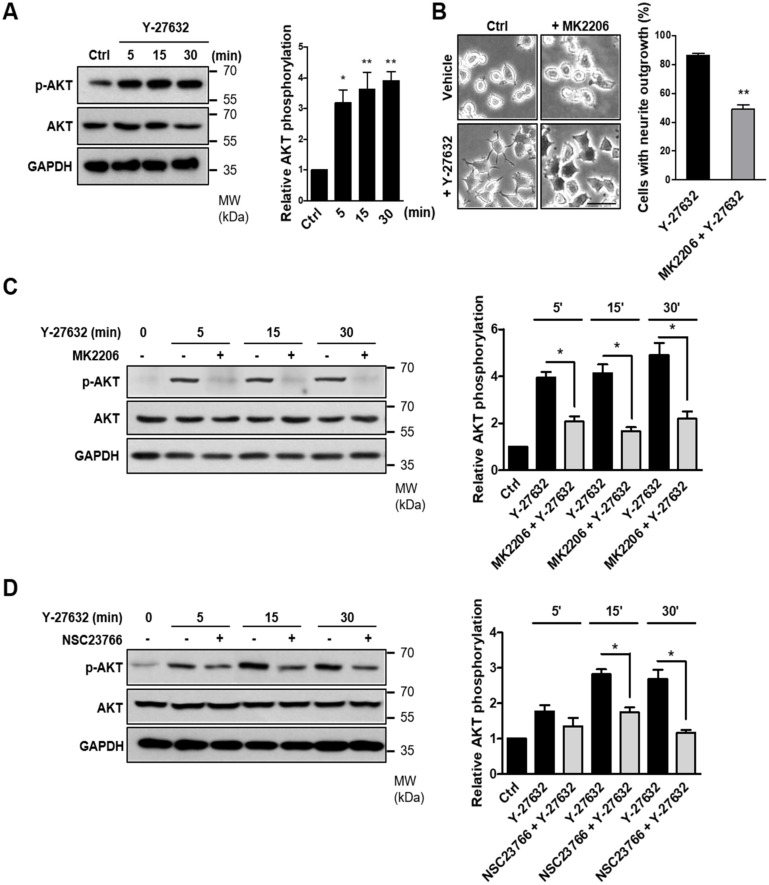
Activation of the Rac1-NOX1-AKT-PAK1 cascade results in neurite outgrowth of PC12 cells. (**A**) PC12 cells were stimulated with 25 μM Y-27632 for the indicated times and cell lysates were then immunoblotted with antibodies against phosphorylated AKT (at Ser473), AKT, and GAPDH. Note that Y-27632 induced the phosphorylation of AKT. (**B**) PC12 cells were stimulated with 25 μM Y-27632 for 2 h in the presence or absence of 5 μM MK2206, an AKT inhibitor. Cells with neurite processes longer than the cell body diameter were counted, and the percentages of neurite-containing cells were determined. Note that the pretreatment of cells with MK2206 significantly inhibited Y-27632-induced neurite outgrowth. Scale bar = 50 µm. Culture images are at a magnification of 20×. (**C**) PC12 cells were treated with 25 μM Y-27632 for the indicated times in the presence or absence of 5 μM MK2206, and the cell lysates were then immunoblotted with antibodies against phosphorylated AKT (at Ser473), AKT, and GAPDH. The p-AKT protein expression level was normalized to that of total AKT and is shown in the right panel. (**D**) PC12 cells were treated with 25 μM Y-27632 for the indicated times in the presence or absence of 60 μM NSC23766, and the cell lysates were then immunoblotted with antibodies against phosphorylated AKT (at Ser473), AKT, and GAPDH. The p-AKT protein expression level was normalized to that of total AKT and is shown in the right panel. (**E**) PC12 cells were treated with 25 μM Y-27632 for the indicated times in the presence or absence of ROS scavengers, such as 5 mM NAC and 500 μM trolox, and the cell lysates were then immunoblotted with antibodies against phosphorylated AKT (at Ser473), AKT, and GAPDH. The p-AKT protein expression level was normalized to that of total AKT and is shown in the right panel. (**F**) PC12 cells were treated with 25 μM Y-27632 for the indicated times in the presence or absence of 10 μM DPI, and the cell lysates were then immunoblotted with antibodies against phosphorylated AKT (at Ser473), AKT, and GAPDH. The p-AKT protein expression level was normalized to that of total AKT and is shown in the right panel. (**G**) PC12 cells transfected with control shRNA or NOX1 shRNA were treated with 25 μM Y-27632 and the cell lysates were then immunoblotted with antibodies against phosphorylated AKT (at Ser473), AKT, and GAPDH. The p-AKT protein expression level was normalized to that of total AKT and is shown in the right panel. (**H**) PC12 cells were treated with 25 μM Y-27632 for the indicated times in the presence or absence of 5 μM MK2206, and the cell lysates were then immunoblotted with antibodies against phosphorylated PAK1 (at Ser199/204), PAK1, and GAPDH. The p-PAK1 protein expression level was normalized to that of total PAK1 and is shown in the right panel. Note that inhibition of AKT by MK2206 decreased Y-27632-induced phosphorylation of PAK1, suggesting that AKT is upstream of PAK1. The data represent the means ± S.E.M. of at least four independent experiments. * *p* < 0.05; ** *p* < 0.01.

**Figure 6 ijms-21-07679-f006:**
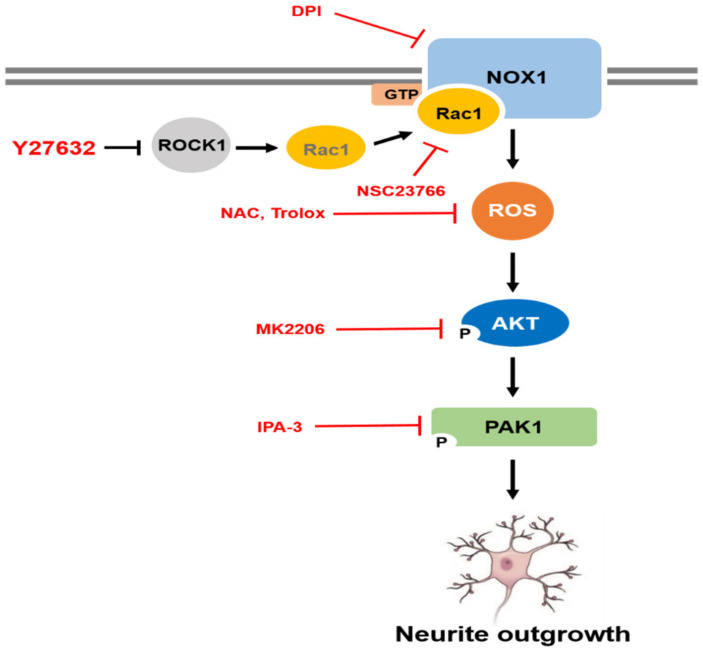
Schematic representation of Y-27632-induced neurite outgrowth in PC12 cells. Y-27632 causes Rac1 activation and leads to NOX1-mediated ROS generation. ROS then subsequently activate AKT and PAK1, which results in neurite outgrowth in PC12 cells.

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
