# Peer review of "Y-27632 Induces Neurite Outgrowth by Activating the NOX1-Mediated AKT and PAK1 Phosphorylation Cascades in PC12 Cells"

_ijms, 2020, doi:10.3390/ijms21207679_

Round 1

Reviewer 1 Report

The manuscript titled “Y-27632 induces neurite outgrowth by activating the 2 NOX1-mediated AKT and PAK1 phosphorylation 3 cascades in PC12 cells” by Park et al. focuses on understanding the molecular mechanism of Y-27 induced neurite outgrowth. There are a few major and minor concerns that need to be addressed before publication.

  1. The PC12 cell line has a natural propensity for neurite outgrowth. How do the authors differentiate the effect of Y-27 treatment? Perhaps a demonstration with a primary neuronal culture would bolster their claims??
  2. Y-27 compound has also been shown to inhibit PCKs. Are there any roles of PCKs in the neurite outgrowth?
  3. Figure quality needs to be improved. They appear somewhat out of focus/ blurry. Labels are not legible. The font sizes are not consistent even within the same figure.
  4. Neurite growth at higher magnification should be shown with neurite growth cone with lamellipodia and filopodia. A more detailed quantification is requested. This is especially true since Rac1 will be modulating the lamellipodia.
  5. To knockdown NOX1, the authors should demonstrate with more than 1 shRNA construct.
  6. What kind of ECM was used for PC12 cells?
  7. This is where I got lost when the authors discussed the involvement of Rac1 mediated neurite outgrowth in Y-27 induced condition. If Rac1 is activating ROCK and Y-27 is a specific inhibitor of ROCK. Then please explain how Y-27 is activating Rac1 and subsequent neurite growth?

Author Response

The manuscript titled “Y-27632 induces neurite outgrowth by activating the 2 NOX1-mediated AKT and PAK1 phosphorylation 3 cascades in PC12 cells” by Park et al. focuses on understanding the molecular mechanism of Y-27 induced neurite outgrowth. There are a few major and minor concerns that need to be addressed before publication.

1. The PC12 cell line has a natural propensity for neurite outgrowth. How do the authors differentiate the effect of Y-27 treatment? Perhaps a demonstration with a primary neuronal culture would bolster their claims??

(Ans.) ThePC12 (pheochromocytoma) cell line is originated from rat chromaffin cells and it has a round shape in normal culture condition. When it is treated with neurite promoting agent such as NGF, the morphology changes into neurite-bearing features. Because of this reason, the PC12 cell line is a very useful cell to monitor and screening of neuritogenic effect of novel chemicals.

2. Y-27 compound has also been shown to inhibit PKCs. Are there any roles of PCKs in the neurite outgrowth?

(Ans.) Although Y-27632 can inhibit PKC inhibition at the high concentration, Y-27632 is highly specific for ROCK inhibition. There is no report that Y-27632-induced PKC inhibition at our experimental concentrations. Various members of the protein kinase C (PKC) isoforms have been reported in mediating NGF-dependent neurite outgrowth of PC12 cells. For example, Phorbol ester-induced depletion of cPKC and nPKC enhances NGF-induced neurite outgrowth without altering aPKC (ζ) levels. Inhibition of PKC-ζ with antisense oligonucleotides prevents NGF-induced neurite outgrowth of PC12 cells. Overexpression of aPKC isoforms in PC12 cells enhances NGF-mediated differentiation along with activation of the transcription factor NF-κB. Thus, the role of PKC in neurite outgrowth in PC12 cells is various and acts in an isoform-specific manner.

3. Figure quality needs to be improved. They appear somewhat out of focus/ blurry. Labels are not legible. The font sizes are not consistent even within the same figure.

(Ans.) According to the reviewer’s suggestion, we checked all the figures and improved resolution. The labels were also adjusted in a consistent size in the revised manuscript.

4. Neurite growth at higher magnification should be shown with neurite growth cone with lamellipodia and filopodia. A more detailed quantification is requested. This is especially true since Rac1 will be modulating the lamellipodia.

(Ans.) According to the reviewer’s suggestion, we have tried to monitor lamellipodia and filopodia at the growth cone. However, it failed due to the limitation of the resolution of our microscopy.

5.To knockdown NOX1, the authors should demonstrate with more than 1 shRNA construct.

(Ans.) We completely agree that we should use more than one shRNA construct. However, the transfection efficiency of PC12 cells was very low and it was very hard to get a stable clone. However, it was clear that our clone is stably knock-downed the endogenous Nox1 expression (Fig.2C).

6. What kind of ECM was used for PC12 cells?

(Ans.) Collagen (100 ug/ml) was used for the attachment and to induce neurite outgrowth. It is now included in the material and methods section in the revised manuscript.

7. This is where I got lost when the authors discussed the involvement of Rac1 mediated neurite outgrowth in Y-27 induced condition. If Rac1 is activating ROCK and Y-27 is a specific inhibitor of ROCK. Then please explain how Y-27 is activating Rac1 and subsequent neurite growth?

(Ans.) Rac1 and RhoA have opposing functions in the process of neurite outgrowth. Rac1 has a neurite promoting effect. In contrast, RhoA has a negative effect on neurite generation. In previous reports, RhoA showed to inhibit Rac1 activation through the ROCK-dependent pathway (Yamaguchi et al., 2001; Takefuji et al, 2007). Consistent with these reports, our result suggested that expression of dominant-negative Rac1 prevented Y-27632 (ROCK inhibitor)-induced neurite outgrowth, indicating that Rac1 is necessary for the Y-27632-induced neurite outgrowth in PC12 cells (Fig. 3B).

Reviewer 2 Report

In this paper by Park et al, the authors highlight mechanisms underlying neurite outgrowth in PC12 cells via the by Y-27632-mediated activation of Rac1/NADPH oxidase (NOX)/AKT/p21-activated kinase 1 (PAK1) signaling cascade.

Overall, the manuscript is informative and produces critical information, however there are some concerns for the authors to address:

The treatment times of Y-27632 vary throughout the manuscript. Based on Fig 1B, the treatment of 2h which is used majority of the times makes sense. However, in Fig 1D, they only treat for 1h and in Figs 3,4,5 the treatment times are less than an 1h for some Western blots. It is okay to show earlier timepoints as signal might be saturated at the longer time points for some assays but state the reason for the different treatment times.

Fig.2F: explain better in text. What is phallodin staining indicative of and how does it corelate to the conclusion that neurite outgrowth is increased in response to Y-27632.

Abbreviation list can be longer. Eg: AKT, GAPDH, NGF

Scale bars are missing in all images. As is the magnification at which the images are taken.

Add mol wt. markers to Western blots.

The Methods section has some missing details:

  1. Section 4.3 should be before 4.2.
  2. What diluent are the inhibitors dissolved in? eg: is it DMSO?
  3. Are the No-inhibitor conditions used in their figures, “a diluent control, eg DMSO control” or “only media”? State.
  4. What is the wavelength at which the CM-H2DCFDA was measured?
  5. State vendor and catalog number for Rac1 expression vectors

Both the Abstract and Discussion have little to no description of the overall translational impact of the study ie which disease state this neurite outgrowth is relevant in, how it would help to develop therapy, etc. Additionally, the Discussion also lacks what the next steps could be for this study, what the limitations are; eg.: only on cell line, no in vivo data etc and how their study is significant despite these limitations. All these points should be accounted for

Author Response

In this paper by Park et al, the authors highlight mechanisms underlying neurite outgrowth in PC12 cells via the by Y-27632-mediated activation of Rac1/NADPH oxidase (NOX)/AKT/p21-activated kinase 1 (PAK1) signaling cascade.

Overall, the manuscript is informative and produces critical information, however, there are some concerns for the authors to address:

The treatment times of Y-27632 vary throughout the manuscript. Based on Fig 1B, the treatment of 2h which is used majority of the times makes sense. However, in Fig 1D, they only treat for 1h and in Figs 3,4,5 the treatment times are less than an 1h for some Western blots. It is okay to show earlier timepoints as signal might be saturated at the longer time points for some assays but state the reason for the different treatment times.

 (Ans.) To detect the morphological features of neurite outgrowth in response to Y-27632 in PC12 cells, the treatment time should be increased for 2 h. However, upstream signaling cascade such as phosphorylation is reached the maximum level at very early times and decreased thereafter in Western blot analysis. Fluorescent-bound ROS stays quite stable than phosphorylation, we can monitor until 1 h under fluorescence microscopy.

Fig.2F: explain better in text. What is phallodin staining indicative of and how does it correlate to the conclusion that neurite outgrowth is increased in response to Y-27632.

(Ans.) Phalloidin is known to bind actin filament. According to the reviewer’s suggestion, it is included in the result section of the revised manuscript as follows. “To monitor neurite outgrowth, Alexa 688-conjugated phalloidin was used as a specific probe for the binding of filamentous actin.”

Abbreviation list can be longer. Eg: AKT, GAPDH, NGF

 (Ans.) According to the reviewer’s suggestion, we added GAPDH, NGF, ROCK, ROS, NAC, and DPI in the revised manuscript. However, AKT is not the form of abbreviation. It is a protein name which is originated from viral oncogene.

Scale bars are missing in all images. As is the magnification at which the images are taken.

 (Ans.) According to the reviewer’s suggestion, scale bars were added throughout the revised manuscript.

Add mol wt. markers to Western blots.

  (Ans.) According to the reviewer’s suggestion, molecular weight markers were added throughout the revised manuscript.

The Methods section has some missing details:

1. Section 4.3 should be before 4.2.

(Ans.) According to the reviewer’s suggestion, section 4.3 is moved before 4.2 in the revised manuscript.

2. What diluent are the inhibitors dissolved in? eg: is it DMSO?

(Ans.) Y-27632, NSC23755, NAC, and trolox were dissolved in ddH2O. IPA-3, DPI, and MK-2206 were dissolved in dimethyl sulfoxide (DMSO). According to the reviewer’s suggestion, it is now included in the revised manuscript.

3. Are the No-inhibitor conditions used in their figures, “a diluent control, eg DMSO control” or “only media”? State.

(Ans.) All chemicals were diluted in culture media to the final concentrations just before use. Culture media was used as a control. According to the reviewer’s suggestion, it is now included in the revised manuscript.

4. What is the wavelength at which the CM-H2DCFDA was measured?

(Ans.) CM-H2DCFDA was measured under a fluorescence microscope at excitation 495 and emission 519 nm wavelength. According to the reviewer’s suggestion, it is now included in the revised manuscript.

5. State vendor and catalog number for Rac1 expression vectors

(Ans.) Rac1 expression vectors were kindly provided by Prof. Jae Hong Kim. (Korea University, Seoul, Korea). According to the reviewer’s suggestion, it is now described in the revised manuscript.

Both the Abstract and Discussion have little to no description of the overall translational impact of the study ie which disease state this neurite outgrowth is relevant in, how it would help to develop therapy, etc. Additionally, the Discussion also lacks what the next steps could be for this study, what the limitations are; eg.: only on cell line, no in vivo data etc and how their study is significant despite these limitations. All these points should be accounted for

(Ans.) According to the reviewer’s comment, we added the following sentences in the introduction part of the revised manuscript (line 49).

“For example, the local application of Y-27632 improved functional recovery in mice with spinal cord injured models (Mueller, Mack, and Teusch 387-398; Sung et al. 29-38). The use of Y-27632 in animal AD models was efficient in lowering the amount of the toxic Ab42 levels, but not in total Ab levels (Zhou et al. 1215-1217).”

In addition, we added the following sentences in the discussion part of the revised manuscript (line 331)

“In support of our notion, Li et al. have recently suggested that the administration of Y-27632 effectively increased the survival rate and behavioral performance of rats from cerebral ischemic injury (Li and Liu 3395-3405). Blocking of ROCK with Y-27632 prevented the initiation of neuropathic pain after peripheral nerve injury in mice (Inoue et al. 712-718). Despite of these reports, the use of Y-27632 in vivo has been limited. Because it metabolized very rapidly after oral administration and its brain penetration was too low to achieve therapeutic levels for the CNS disease. Thus, the intensive investigation will be needed to evaluate the efficacy of Y-27632 before the application of human therapeutics.

Y-27632 has been extensively studied in a variety of cellular models. In accordance with these reports, our study reported the neurite-promoting effect of Y-27632 and suggested the underlying molecular signaling cascade in PC12 cell lines. Due to the limited studies using in vitro system, further studies are expected to confirm whether our hypothesis can be applied to the primary neurons and to the diverse in vivo animal models.”

Round 2

Reviewer 1 Report

The authors have addressed some of the initial concerns I had. But there are still a few that need to be addressed before publication.

1. It would be nice to demonstrate that PKCs are not affected by Y27 compound at the lower concentration used here. At least, The authors need to discuss and add a reference to show that at low concentrations, Y27 compound does not affect PKCs.

2. Knockdown of NOX1 with 1 shRNA construct is not acceptable. They need to provide justification and discuss in the manuscript why using 1 shRNA construct can be a limiting factor for interpreting the data. Hard to transfect is not really a justification.

Author Response

The authors have addressed some of the initial concerns I had. But there are still a few that need to be addressed before publication.

  1. It would be nice to demonstrate that PKCs are not affected by Y27 compound at the lower concentration used here. At least, The authors need to discuss and add a reference to show that at low concentrations, Y27 compound does not affect PKCs.

(Ans.) According to the reviewer’s suggestion, the reference regarding the affinity results of Y-27632 toward different kinases was included in the revised introduction part as follows. “Its affinity for ROCK is over 200 times or 2,000 times higher than that for conventional protein kinase C (PKC) or cAMP-dependent protein kinase (PKA), or myosin light-chain kinase (MLCK), respectively [12].”

  1. Knockdown of NOX1 with 1 shRNA construct is not acceptable. They need to provide justification and discuss in the manuscript why using 1 shRNA construct can be a limiting factor for interpreting the data. Hard to transfect is not really a justification.

(Ans.) We used commercially available NOX1 shRNA plasmid (Santa Cruz, SC-156079-SH). According to the manufacture’s indication, the product is a pool of 3 target-specific plasmids each encoding 19-25 nt (plus hairpin) shRNAs designed to knock down NOX1 expression. We added this in the revised manuscript as follows. “To further evaluate the endogenous role of NOX1 in Y-27632-induced neurite outgrowth, we generated the stable NOX1-knockdown cells using a pool of 3 different target-specific shRNA expressing plasmids. The effect of Y-27632 on the neurite outgrowth was consistently perturbed in the stable NOX1-knockdown cells. However, further studies will be needed to confirm the effect of NOX1-knockdown on the Y-27632-induced neurite outgrowth in PC12 cells using different shRNAs.”